# Lattice Distortion, Amorphization and Wear Resistance of Carbon-Doped SUS304 by Laser Ablation

**DOI:** 10.3390/ma15165764

**Published:** 2022-08-20

**Authors:** Seonghoon Kim, Taewoo Kim, Eunpyo Hong, Heesoo Lee

**Affiliations:** 1School of Materials Science and Engineering, Pusan National University, Busan 46241, Korea; 2System & Energy Division Material Technology Center, Korea Testing Laboratory, Seoul 08389, Korea; 3Mechanical & Material Technology Center, Korea Testing Laboratory, Jinju 52852, Korea

**Keywords:** carbon-doped SUS304, variation of laser output, S-phase, amorphous carbon, friction coefficient

## Abstract

Lattice distortion and amorphization of carbon-doped SUS304 by variation of the laser output were investigated in terms of phase formation and the bonding state. The laser output was changed by 10% in the range of 60% to 100% after covering the SUS304 with carbon paste. A graphite peak and expanded austenite (S-phase) peak were observed in the carbon-doped SUS304, and Rietveld refinement was performed to identify the lattice distortion. The lattice constant of SUS304 was initially 3.612 Å, but expansion lattice distortion occurred in the carbon-doped SUS304 as a result of the S phase formation and carbon doping, and the lattice constant increased to 3.964 Å (100% laser output). X-ray photoelectron spectroscopy analysis for the bonding state of the carbon-doped SUS304 showed that the sp^2^/sp^3^ ratio decreased from 3.21 (70% laser output) to 2.52 (100% laser output). The residual stress in the lattice was accumulated due to carbon doping by high thermal energy, which resulted in the formation of amorphous carbon. The bonding environment was represented by the I_D_/I_G_ ratio using Raman analysis, and it increased from 0.55 (70% laser output) to 1.68 (100% laser output). During microstructure analysis of the carbon-doped SUS304, disordered structures by amorphization were observed in the carbon-doped SUS304 by the greater than 90% laser output. The amorphous carbon filled the lattice grains or voids to lubricate the surface, which improved the friction coefficient and wear rate from 0.23 and 7.63 mm^3^(Nm)^−1^10^−6^ to 0.09 and 1.43 mm^3^(Nm)^−1^10^−6^, respectively.

## 1. Introduction

SUS304, an austenitic stainless steel alloy containing Ni and Cr, has excellent corrosion resistance and thermal stability. It has been used in various industries such as automobile, aerospace, and chemical plants. However, its relatively low surface hardness is susceptible to wear, and pitting corrosion occurs in a seawater atmosphere [1,2]. Some studies have been conducted on surface-hardening mechanisms, such as carburization, nitriding, and carbonitriding. [3,4]. Baranowska et al. discussed microstructural and morphological changes in low-temperature gas carbonitrided layers on austenitic stainless steel depending on the process parameters. [5]. You et al. adjusted mechanical properties by setting the atomistic diffusion mechanism of rare earth carburizing and nitriding on iron-based alloy [6].

Carbon diffuses on the surface of low-carbon steel and forms a high-carbon alloy layer during carburization in SUS304, improving the mechanical properties of the surface [7,8]. Low-temperature plasma carburizing is a representative metal carburizing process that enhances its hardness and fatigue properties by over-dissolving carbon in the austenite lattice without precipitation of CrN and CrC [9,10,11]. However, when the low-temperature plasma carburizing process is performed, the improvement of the friction coefficient is insufficient. Therefore, methods for improving the wear resistance of austenitic stainless steel have been actively studied [12,13].

Laser carburization is a surface modification process used in various fields to improve the mechanical properties of materials. The thermal energy of the laser is the energy source for carbon diffusion when the carbon is doped by the laser [14,15]. When the laser power is increased, amorphous carbon is formed in the austenite lattice due to the accumulation of heat and stress, along with carbon doping by diffusion [16,17]. Amorphous carbon (a-C) is chemically inert and can act as solid lubricant, preventing crack propagation due to hybridization in the austenite lattice and an improved hardness. Some studies on the amorphization of steel containing unusual alloying elements such as Y, Zr, and B have been conducted. Research on the amorphization of stainless steel has been also spotlighted in terms of friction, wear, and corrosion [18,19,20]. Yang et al. discussed the mechanical and tribological properties of the catalytic growth of diamond-like carbon on Fe3C containing a carburized layer. [21]. Dalibón et al. confirmed the chemical microstructure and tribological properties of thick and soft DLC coatings deposited on plasma-nitrided austenitic stainless steel. [22].

We investigated the lattice distortion and amorphization of carbon-doped SUS304 by variation of laser output. The phase formation and lattice constant were analyzed using X-ray diffraction analysis and the Rietveld refinement method. X-ray photoelectron spectroscopy, Raman spectroscopy, and transmission electron microscopy were performed to confirm the bonding environment of carbon by laser output. The variation in the wear resistance was identified by measuring the friction coefficient and the wear rate of carbon-doped SUS304 using the ball-on-disc test.

## 2. Materials and Methods

The surface of SUS304 was cleaned using ultrasonification and ethanol to improve the adhesion of the carbon paste by removing any contamination before deposition. Carbon paste was mixed with graphite powder (20 µm) and polyvinylidene fluoride (PVDF) at a wt% ratio of 9:1, and the mixed powder and solvent (1-Methyl-2-pyrrolidinone) were mixed in a ratio of 1:2 wt% to control the viscosity of the carbon paste. The carbon paste was covered on SUS304 using screen printing and dried at 80 °C to the improve adhesion. Carburization on the surface of SUS304 was performed using an Nd:YAG pulsed laser ablation system (LSX-213) (Figure 1). The laser was irradiated 10 times with a wavelength of 213 nm, a frequency of 10 Hz, and spot distance of 200 µm and 250 µm. The laser output was changed to 60% (2.81 mJ), 70% (3.26 mJ), 80% (3.71 mJ), 90% (4.14 mJ), and 100% (4.61 mJ), and the carbon paste remaining on the surface was removed with an ultrasonic wave, acetone, and ethanol after laser carburization.

The crystallographic change was analyzed using an X-ray diffractometer (Rigaku, Ultima-IV, Tokyo, Japan) with a Cu target at 0.02°/2θ step from 20° to 80° at 1° per minute. The gradient of diagram (variation in 2θ as a function of sin2Ψ) required for the calculation of the residual stress was obtained by varying the stress on a specific plane by changing the Ψ angle from 0° to 50°. The lattice constant was calculated by Rietveld refinement through X’pert High Score (Panalytical, Almelo, The Netherlands). X-ray photoelectron spectroscopy (ESCALAB250, VG Scientific, Waltham, MA, USA) was carried out for the chemical characterization of the samples with an MXR1 gun 400 µm-15 kV spectrometer with an Al Kα source at the KBSi Busan Center. The bonding configuration analysis was performed by Raman spectroscopy (Vertex 80 V, Bruker, Billerica, MA, USA). For the microstructure analysis, the specimens were prepared using a Focused Ion Beam (FIB, Helios G4 UC, Thermo Fisher Scientific, Waltham, MA, USA), and the cross-section of the coatings was observed with TEM (Cs-TEM, Titan Themis Z, Thermo Fisher Scientific, Waltham, MA, USA). The friction coefficients and wear rate were measured by the ISO 20808 standard test method using a ball-on-disc system (NEO-Tribo Multi-Purpose Wear Test System, Neoplus Inc., Dajeon, Korea) [23].

## 3. Results and Discussion

Figure 2 shows the structure analysis of the carbon-doped SUS304 by variation of laser output. At 60% laser output, the phase formation and peak shift could not be observed by low thermal energy. From 70% laser output, the formation of the expanded austenite (S-phase) by carburization was confirmed through the shift in the austenite (111) peak to a low angle with the graphite peak. Table 1 compared the lattice distortion after carbon doping by Rietveld refinement, and the lattice constant increased with the laser output. It is considered that the expansion of the lattice distortion occurred due to carbon doping in the faced-centered cubic (FCC) structure of SUS304 and the formation of metastable supersaturated expanded austenite [24,25].

Figure 3 shows the result of the XPS C 1s diffraction pattern, with the variation in the bonding state of carbon-doped SUS304. The lowest energy peak (283.0 eV) corresponded to the carbide (Fe-C or Cr-C) peak, and 284.5 eV and 286.2 eV represented sp^2^ and sp^3^ bonds, respectively [26,27]. The 290.4 eV revealed a CF bond with fluorine in the carbon paste. At a low laser output, the sp^2^ bond with a low bonding strength was dominant, so carbide formation continued to increase corresponding to the increase in laser output. The solid solution limit of carbon was exceeded after reaching 90% laser output, and amorphous carbon started to form in the vacancy or gain of the carbon-doped SUS304. As a result of calculating the sp^2^/sp^3^ ratio, it showed 3.21 at 70% laser output, while it gradually decreased as the laser output increased, representing 2.67 and 2.52 at 90 and 100% laser output, respectively. It was revealed that amorphous carbon was formed due to the accumulation of residual stress forming inside the lattice by high thermal energy and carbon doping [28,29]. However, in 100% laser output, carbon hydroxides were observed by the inflow of excessive thermal energy, and deterioration occurred in the SUS304 (Table 2).

The bonding environment of the carbon-doped SUS304 was analyzed using Raman spectroscopy. The D band (1320 to 1350 cm^−1^) in the Raman spectrum indicates a defect or disorder and corresponds to atoms moving in radial directions in the lattice. The G band (1500 to 1600 cm^−1^) means in-plane vibration of sp^2^ hybridized bonds and corresponds to neighboring atoms moving in opposite directions perpendicular to the plane of the lattice [30,31]. In Figure 4, the peak intensity of the D band increased with the variation in laser output, but the peak intensity of the G band showed the opposite flow, which decreased with the laser output. Table 3 reveals the result of calculating the I_D_/I_G_ ratio of carbon-doped SUS304, and it was increased from 0.68 (70% laser output) with carbon doping and indicated a maximum value of 1.68 at 100% laser output. It was represented that the sp^2^ configuration predominated at the low laser output and was preferred for bond formation as it was more stable. When the laser output was changed from 80% to 90%, the I_D_/I_G_ ratio increased by 45%. Amorphization of carbon-doped SUS304 occurred due to the graphitization and the increase in defects with disordering.

Figure 5 represents a cross-section in TEM images to understand the microstructure of the SUS304 and carbon-doped SUS304 by laser carburization. Figure 5b–e show the microstructure of carbon-doped SUS304 at 70%, 80%, 90%, and 100% laser output, respectively. In Figure 5b,c, the carbon-doped SUS304 has an ordered lattice structure due to the low laser output. However, in Figure 5d,e, a disordered lattice structure appeared due to high laser output, and it was confirmed that amorphization occurred in the carbon-doped SUS304 corresponding the Raman analysis.

Figure 6 reveals the results of the friction coefficient and wear rate of the carbon-doped SUS304 using a ball-on-disc tester to confirm the variation in wear resistance before and after laser carburization. The friction coefficient decreased from 0.23 (60%) to 0.2 (80%), 0.09 (90%), and 0.12 (100%) respectively, and the wear rate was 7.63 mm^3^(Nm)^−1^10^−6^ (70%), 6.76 mm^3^(Nm)^−1^10^−6^ (80%), 2.07 mm^3^(Nm)^−1^10^−6^ (90%), and 2.45 mm^3^(Nm)^−1^10^−6^ (100%). In the low range of the laser output (70, 80%), it is judged that the tribological properties decreased due to the distortion of the lattice due to the formation of S-phase in carbon-doped SUS304 and the improvement in the residual stress and hardness. However, in the high range of the laser output (90, 100%), it is considered that the tribological properties were reduced because the formed amorphous carbon filled the grains or voids in the lattice, and the surface was lubricated. The decrease in wear resistance at 100% laser output compared to 90% laser output is due to the formation of carbon hydroxides by the inflow of excessive thermal energy [32,33].

## 4. Conclusions

We investigated the lattice distortion, amorphization, and wear resistance of carbon-doped SUS304 by varying the laser output. The laser output was increased by 10% in the range from 60% to 100%. The diffraction peaks of graphite and the S-phase were formed after carburization, and the lattice constant increased from 3.612 Å to 3.964 Å, indicating the expansion lattice distortion. At a low laser output, the sp^2^ bond with low bonding strength was dominant, but the solid solution limit of carbon was exceeded after reaching 90% laser output, and amorphous carbon began to form in the vacancy or grain of the carbon-doped SUS304. The I_D_/I_G_ ratio calculated with Raman analysis increased from 0.68 (70% laser output) to 1.68 (100% laser output), which indicated that amorphization of carbon occurred due to the accumulation of the thermal energy and residual stress formed in the carbon-doped SUS304. For TEM analysis, the microstructure of the carbon-doped SUS304 became disordered as the laser output increased. The friction coefficient and wear rate were measured to confirm the wear resistance and improved from 0.23 and 7.63 mm^3^(Nm)^−1^10^−6^ to 0.09 and 1.43 mm^3^(Nm)^−1^10^−6^, respectively.

## Figures and Tables

**Figure 1 materials-15-05764-f001:**
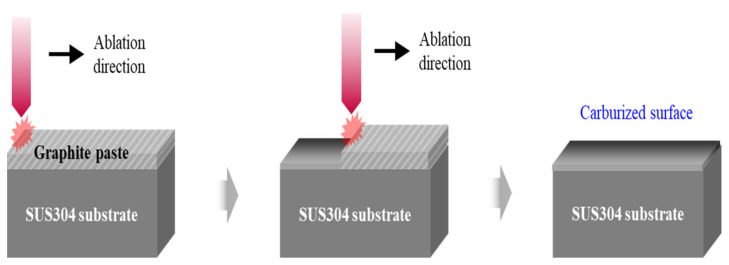
Schematic of laser carburization on SUS304.

**Figure 2 materials-15-05764-f002:**
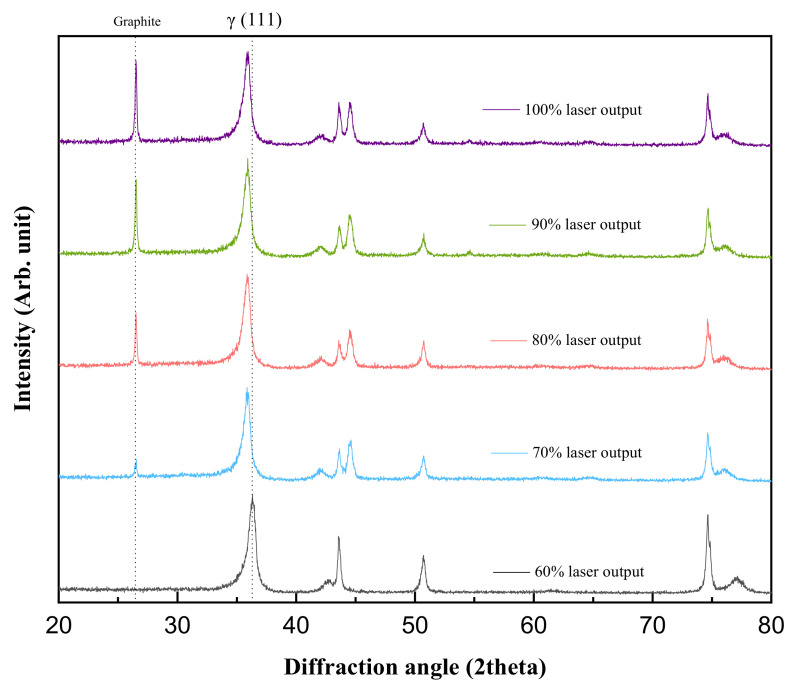
XRD patterns of carbon-doped SUS304 peaks by laser output.

**Figure 3 materials-15-05764-f003:**
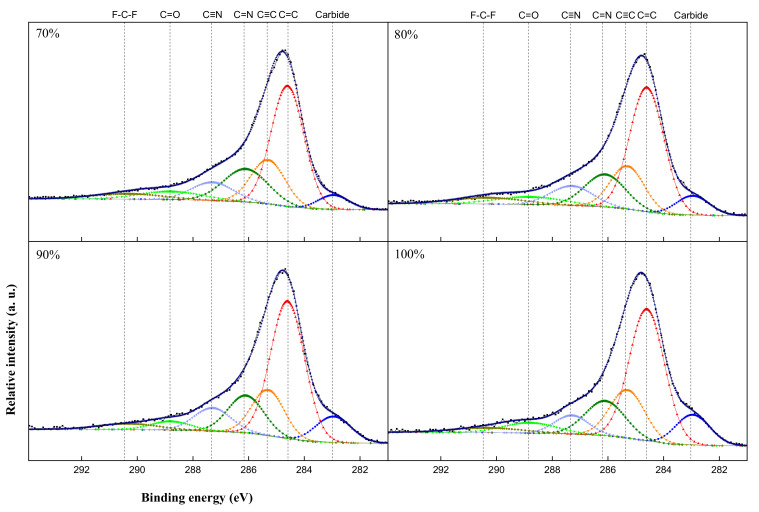
XPS C 1s spectra of carbon-doped SUS304 by laser output.

**Figure 4 materials-15-05764-f004:**
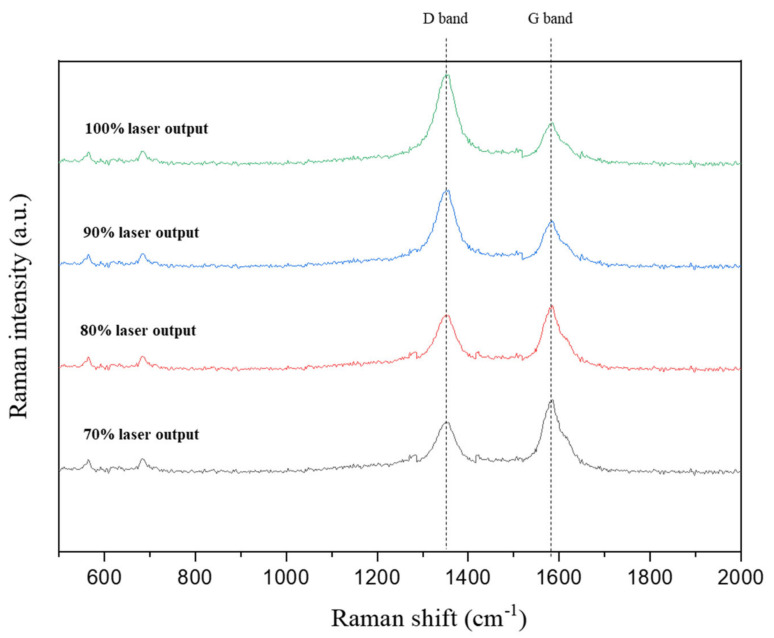
Raman spectra of carbon-doped SUS304 by laser output.

**Figure 5 materials-15-05764-f005:**
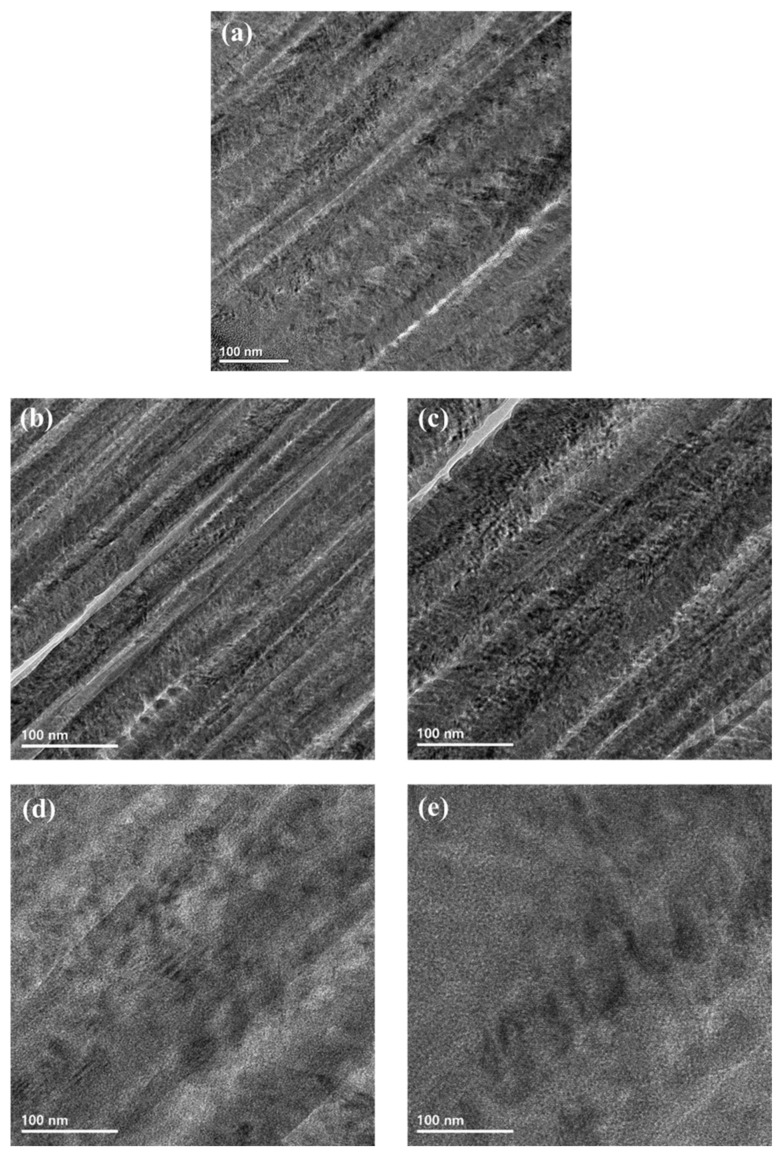
Cross-section TEM images of SUS304 and carbon-doped SUS304: (**a**) as-deposited, (**b**) 70% laser output, (**c**) 80% laser output, (**d**) 90% laser output, and (**e**) 100% laser output.

**Figure 6 materials-15-05764-f006:**
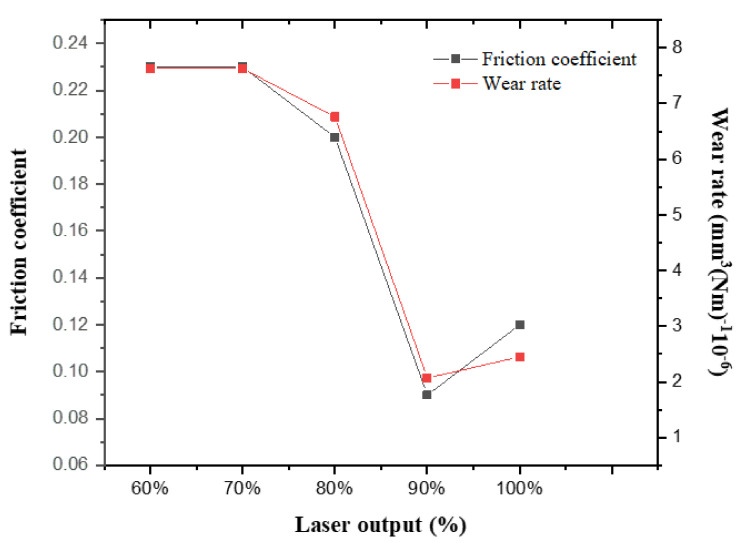
Friction coefficient and wear rate of as-deposited SUS304 and carbon-doped SUS304 by laser output.

**Table 1 materials-15-05764-t001:** Lattice constant before and after the carbon doping through the Rietveld refinement.

Laser Output	Lattice Constant
As-deposited	3.612 Å
60%	3.612 Å
70%	3.674 Å
80%	3.749 Å
90%	3.915 Å
100%	3.964 Å

**Table 2 materials-15-05764-t002:** sp^2^/sp^3^ ratio of carbon-doped SUS304 by laser output.

Laser Output	sp^2^/sp^3^ Ratio
70%	3.21
80%	3.14
90%	2.67
100%	2.52

**Table 3 materials-15-05764-t003:** I_D_/I_G_ of carbon-doped SUS304 by laser output.

LASER Output	I_D_/I_G_
70%	0.68
80%	0.85
90%	1.24
100%	1.68

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
