# Peer review of "Lattice Distortion, Amorphization and Wear Resistance of Carbon-Doped SUS304 by Laser Ablation"

_materials, 2022, doi:10.3390/ma15165764_

Round 1
Reviewer 1 Report
---Please see the attached review report for better formatting ---
Review report
REF: materials-1858626
Title: Lattice distortion, amorphization, and wear resistance of carbon-doped SUS304 by laser ablation
General comment:
This manuscript shows the lattice distortion and amorphization of a carbon-doped steel alloy. The laser-induced stress was studied in a systematic range (10-100%). Lattice distortion is studied in terms of the lattice constant, and disordered structures were observed given rise by the laser-induced thermal stresses. As a result of the carburization, the wear resistance is improved significantly. The manuscript is well written, and the results are clearly presented with full information. However, prior to the final decision to be made, I suggest the authors should consider addressing the following issues, especially the mechanism for the observed phenomenon with discussion to the literature
Please note all references mentioned in the comments below are for guidance only.
Major concerns:
The lattice distortion was measured in terms of lattice constant. The audience may wonder the lattice distortion is relevant to the lattice rotation that has been observed due to frictional surface stress (e.g. http://doi.org/10.1021/acsami.6b04035 , http://10.1016/j.ijplas.2021.102942) .
Figure 4 shows beautiful cross-section TEM of the samples subject to different levels of laser output. Therefore, it would of great help that the authors extract the dislocation information that is given by the induced stress. A comparison among them would show insightful understanding of the surface deformation. This would also help the understanding of the improvement of the wear resistance, which has been shown to be related to the dislocation activities (e.g. http://doi.org/10.3390/ma14216511 ). By the way, the S4 figure is not available for reviewing so it’s hard to assess.
When discussion the wear resistance, the authors should be cautious about the definition of COF, which has been shown to be contact-size dependent. Since the authors have shown the microstructure and its change in Figure 1 and Figure 4, it would be more accurate to specify or discuss the microstructure-sensitive COF.
Suggestion:
Revision
Author Response
Thank you for compliments and comments to the manuscript. Your sincere reviews have been very helpful to improve the quality of this paper.
Below is our reply to your comments.
[Reviewer’s comments]
Q 1) This manuscript shows the lattice distortion and amorphization of a carbon-doped steel alloy. The laser-induced stress was studied in a systematic range (10-100%). Lattice distortion is studied in terms of the lattice constant, and disordered structures were observed given rise by the laser-induced thermal stresses. As a result of the carburization, the wear resistance is improved significantly. The manuscript is well written, and the results are clearly presented with full information. However, prior to the final decision to be made, I suggest the authors should consider addressing the following issues, especially the mechanism for the observed phenomenon with discussion to the literature.
Please note all references mentioned in the comments below are for guidance only.You will see that a number of general and specific points are mentioned which necessitate extensive rewriting of the paper. You will see that a number of general and specific points are mentioned which necessitate extensive rewriting of the paper.
Answer) Thank you for the comment. We have faithfully revised the results and discussion part according to the reviewer’s comments. Also, we fixed the reference with the guidance for author.
-------------------------------------------------------------------------------------
Q 2) The lattice distortion was measured in terms of lattice constant. The audience may wonder the lattice distortion is relevant to the lattice rotation that has been observed due to frictional surface stress
(e.g. http://doi.org/10.1021/acsami.6b04035 , http://10.1016/j.ijplas.2021.102942).
Answer) Thank you for the kind comments. The lattice distortion in the manuscript is about the variation in lattice constant due to carbon doping when carbon paste covered on SUS304 is irradiated with a laser. Carbon doping in SUS304 can be expressed as a mechanism that occurs when thermal energy is accumulated by the laser irradiated to the carbon paste. So, we thought that the effect lattice distortion and frictional surface stress is negligible.
-------------------------------------------------------------------------------------
Q 3) Figure 4 shows beautiful cross-section TEM of the samples subject to different levels of laser output. Therefore, it would of great help that the authors extract the dislocation information that is given by the induced stress. A comparison among them would show insightful understanding of the surface deformation. This would also help the understanding of the improvement of the wear resistance, which has been shown to be related to the dislocation activities (e.g. http://doi.org/10.3390/ma14216511 ). By the way, the S4 figure is not available for reviewing so it’s hard to assess.
Answer) We fixed the S4 figure to Figure 4 by reviewer’s comment.
-------------------------------------------------------------------------------------
Q 4) When discussion the wear resistance, the authors should be cautious about the definition of COF, which has been shown to be contact-size dependent. Since the authors have shown the microstructure and its change in Figure 1 and Figure 4, it would be more accurate to specify or discuss the microstructure-sensitive COF.
Answer) Thank you for the comment. Regarding the friction coefficient and ball-on disc test, the test was conducted according to the ISO 20808 standard in order to minimize the effect of experimental variables such as contact-size dependent.
Reviewer 2 Report
Journal: Materials (ISSN 1996-1944)
Manuscript ID: materials-1858626
Type: Article
Title: Lattice distortion, amorphization, and wear resistance of carbon-doped SUS304 by laser ablation
Recommendation: Major revisions
This paper reports the lattice distortion, amorphization and wear resistance investigation of the carbon-doped SUS304 subjected by laser ablation (laser-induced thermal stress), which provides an interesting contribution on the metallurgy of this type of steel grades.
The manuscript was attractive, well organize and clear. However, it was not sufficient novelty. Authors only report their results and discussed about them, and made conclusions without new findings.
Therefore, it presents a number of shortcomings that must be addressed before the paper can be considered for publication, as listed below:
1. It must be declared the purposes, significances and novelty of the study in “Abstract” and “Introduction”, rather than simply reporting the experimental results.
2. References were old, average years of references was less than 2010 (Even, 1973 was ignored). Therefore, authors should replace some old references reasonably by the latest literatures to introduce the state-of-the-art advances in austenitic stainless steels in the revised manuscript.
3. In the Keywords, the “Carbon hybridization” may be a mistake. Please check the full text carefully to avoid similar mistakes.
4. The introduction must be improved sufficient background and include all relevant references. For example, in line 51-52: “Furthermore, research on the amorphization of stainless steel has been spotlighted in terms of friction, wear, and corrosion [17-19]”, the details of the researches should be provided.
5. How do the lattice constants were figured out by the Rietveld refinement? The ways and details of the Rietveld refinement should be explain.
6. In Results and discussion, the lattice constant of the SUS304 without the carbon doping should be explain by citing previous articles. Moreover, the energy values of different bonds should be declared by citing some papers.
7. The Fig 4(a), 4(b), 4(c), and 4(d) were not identified in the Figure 4. And in line 131, the “Figure S4” may be a mistake.
8. In this paper, the expression form needs to be unified. For example, there are different expression forms in line 88 (Figure 1), line 100 (Fig. 2), line 132 (Fig 4), line 114 (ID/IG), line 125 (ID/IG) etc. It is strongly advised that the authors take all necessary measures to improve language quality and style.
9. It is necessary to replace the TEM images with the HRTEM (High Resolution Transmission Electron Microscope) images. And the microstructure of SUS304 without the carbon doping should be provided.
10. In line 110-111, what is the original composition of carbon hydroxides in SUS304? Where are H and O from? In line 151-153, what is the mechanism of the formation of carbon hydroxides in SUS304 affecting the wear resistance? Please explain briefly the mechanism.
11. The conclusion should be amended as appropriate, which can be divided into multiple sections to explain. It is necessary to write in simple and clear.
Author Response
Thank you for compliments and comments to the manuscript. Your sincere reviews have been very helpful to improve the quality of this paper.
Below is our reply to your comments.
[Reviewer’s comments]
Q 1) It must be declared the purposes, significances and novelty of the study in “Abstract” and “Introduction”, rather than simply reporting the experimental results.
You will see that a number of general and specific points are mentioned which necessitate extensive rewriting of the paper. You will see that a number of general and specific points are mentioned which necessitate extensive rewriting of the paper.
Answer) Thank you for the comment. We have revised the abstract and introduction to emphasize the novelty, major finding, and conclusion according to the reviewer’s comments.
-------------------------------------------------------------------------------------
Q 2) References were old, average years of references was less than 2010 (Even, 1973 was ignored). Therefore, authors should replace some old references reasonably by the latest literatures to introduce the state-of-the-art advances in austenitic stainless steels in the revised manuscript.
Answer) As the reviewer mentioned, we modified some of the references related to the state-of-the-art advances in austenitic stainless steels.
-------------------------------------------------------------------------------------
Q 3) In the Keywords, the “Carbon hybridization” may be a mistake. Please check the full text carefully to avoid similar mistakes.
Answer) Thank you for the comments. We used the ‘Carbon hybridization’ to describe the changes in sp2 bond, sp3 bond, and amorphization that occur when carbon is doped into the austenitic stainless steel. So, we modified the keyword from ‘carbon hybridization’ to ‘amorphization’.
-------------------------------------------------------------------------------------
Q 4) The introduction must be improved sufficient background and include all relevant references. For example, in line 51-52: “Furthermore, research on the amorphization of stainless steel has been spotlighted in terms of friction, wear, and corrosion [17-19]”, the details of the researches should be provided.
Answer) Thank you for the comments. We have faithfully revised the introduction section according to the number of general and specific points to make it suitable.
-------------------------------------------------------------------------------------
Q 5) How do the lattice constants were figured out by the Rietveld refinement? The ways and details of the Rietveld refinement should be explain.
Answer) Thank you for the comments. We used the sin2Ψ method to figure out the Rietveld refinement, and also added the relevant information in the Materials & Methods part.
-------------------------------------------------------------------------------------
Q 6) In Results and discussion, the lattice constant of the SUS304 without the carbon doping should be explain by citing previous articles. Moreover, the energy values of different bonds should be declared by citing some papers.
Answer) As the reviewer mentioned, we added the lattice constant of the SUS304 without the carbon doping, and noted the reference regarding energy values of different bonds.
-------------------------------------------------------------------------------------
Q 7) The Fig 4(a), 4(b), 4(c), and 4(d) were not identified in the Figure 4. And in line 131, the “Figure S4” may be a mistake.
Answer) Thank you for the comments. We modified the Fig 4(a), 4(b), 4(c), and 4(d) so that you can see it, and replaced “Figure S4” in the line 131 to the “Figure 4”
-------------------------------------------------------------------------------------
Q 8) In this paper, the expression form needs to be unified. For example, there are different expression forms in line 88 (Figure 1), line 100 (Fig. 2), line 132 (Fig 4), line 114 (ID/IG), line 125 (ID/IG) etc. It is strongly advised that the authors take all necessary measures to improve language quality and style.
Answer) Thank you for the comment. We have revised the entire manuscript for the unity of the expression form.
-------------------------------------------------------------------------------------
Q 9) It is necessary to replace the TEM images with the HRTEM (High Resolution Transmission Electron Microscope) images. And the microstructure of SUS304 without the carbon doping should be provided.
Answer) Thank you for the imformative comment. In reality, changing the TEM image to an HR-TEM image in this paper is not easy due to a number of difficulties, so we improved the resolution(dpi). We will make sure to reflect this in future research, and the TEM image of SUS304 without carbon doping has been added to the text.
-------------------------------------------------------------------------------------
Q 10) In line 110-111, what is the original composition of carbon hydroxides in SUS304? Where are H and O from? In line 151-153, what is the mechanism of the formation of carbon hydroxides in SUS304 affecting the wear resistance? Please explain briefly the mechanism.
Answer) Thank you for the comments. We covered carbon paste on SUS304, irradiated with pulsed laser to dope the carbon, and then remove the carbon paste using ultrasonic wave, acetone, and ethanol. When we irradiated the specimen with 100% laser-induced thermal energy, it is judged that the specimen was partially deteriorated, and carbon hydroxide was generated during the cleaning process or exposure to the atmosphere.
-------------------------------------------------------------------------------------
Q 10) The conclusion should be amended as appropriate, which can be divided into multiple sections to explain. It is necessary to write in simple and clear.
Answer)
Thank you for the comment. We have revised the conclusion for conciseness according to the reviewer’s comments.
Reviewer 3 Report
Recommendation: Minor revisions needed as noted.
In this manuscript, the authors presented a laser ablation processing method that can result in higher lattice constant and wear resistance for carbon-doped stainless steel SUS304. Experimental materials and methods were presented, along with the information and results needed to support the authors’ conclusion.
Overall the paper is well written. However, there are a few pieces of missing information, and several minor changes were suggested:
1. Line 64: ‘…at a ratio of 9:1’ is this a weight ratio? Should be specified.
2. Line 64 & 65: ‘1-Methyl-2-pyrrolidinone was used as a solvent to adjust the viscosity 64 of the paste.’ The amount of solvent, or the concentration should be mentioned, and it would be even better that the authors can provide some viscosity measurement results.
3. Line 69: The output energy should be specified.
4. Line 125: Is the laser-induced thermal stress change linear proportionally with the laser output (i.e. is laser output increasing from 80% to 90% shown in Table 3 equivalent to 10% change in laser induced thermal stress)?
5. It will be helpful if the authors can also present a few photos/ schematics of the experimental setup and the test samples to demonstrate their work.
Author Response
Thank you for compliments and comments to the manuscript. Your sincere reviews have been very helpful to improve the quality of this paper.
Below is our reply to your comments.
[Reviewer’s comments]
Q 1) Line 64: ‘…at a ratio of 9:1’ is this a weight ratio? Should be specified.You will see that a number of general and specific points are mentioned which necessitate extensive rewriting of the paper. You will see that a number of general and specific points are mentioned which necessitate extensive rewriting of the paper.
Q 2) Line 64 & 65: ‘1-Methyl-2-pyrrolidinone was used as a solvent to adjust the viscosity 64 of the paste.’ The amount of solvent, or the concentration should be mentioned, and it would be even better that the authors can provide some viscosity measurement results.
Answer) Thanks you for the comments. We revised the Materials & Methods part as follows.
- Before
“Carbon paste was mixed with graphite powder (20 ㎛) and polyvinylidene fluoride (PVDF) at a ratio of 9:1, and 1-Methyl-2-pyrrolidinone was used as a solvent to adjust the viscosity of the paste.”
- After
“Carbon paste was mixed with graphite powder (20 ㎛) and polyvinylidene fluoride (PVDF) at a wt% ratio of 9:1, and the mixed powder and solvent (1-Methyl-2-pyrrolidinone) were mixed in a ratio of 1:2 wt% to control the viscosity of carbon paste.”
-------------------------------------------------------------------------------------
Q 3) Line 69: The output energy should be specified.
Answer) Thank you for the comments. We added the laser energy equivalent to the laser output in line 69.
-------------------------------------------------------------------------------------
Q 4) Line 125: Is the laser-induced thermal stress change linear proportionally with the laser output (i.e. is laser output increasing from 80% to 90% shown in Table 3 equivalent to 10% change in laser induced thermal stress)?
Answer) Thank you for the comment. We used laser-induced thermal stress and laser output interchangeably. So, the word has been unified to prevent misinterpretation of the reader.
-------------------------------------------------------------------------------------
Q 5) It will be helpful if the authors can also present a few photos/ schematics of the experimental setup and the test samples to demonstrate their work.
Answer) Thank you for the kind comments. We added the schematic of laser carburization on SUS304 in Materials & Methods part.
Round 2
Reviewer 2 Report
Journal: Materials (ISSN 1996-1944)
Manuscript ID: materials-1858626
Type: Article
Title: Lattice distortion, amorphization, and wear resistance of carbon-doped SUS304 by laser ablation
Recommendation: Accept